# Aneurysmal Bone Cyst of the Pelvis in Children and Adolescents: Effectiveness of Surgical Treatment with Curettage, Cryotherapy and Bone Grafting

**DOI:** 10.3390/healthcare11192658

**Published:** 2023-09-30

**Authors:** Lorenzo Andreani, Edoardo Ipponi, Elena Serrano, Silvia De Franco, Martina Cordoni, Elena Bechini, Antonio D’Arienzo, Paolo Domenico Parchi

**Affiliations:** Department of Orthopedics and Trauma Surgery, University of Pisa, 56126 Pisa, Italy; lorenzo.andreani.unipi@gmail.com (L.A.); e.serrano@studenti.unipi.it (E.S.); defranco.silvia@gmail.com (S.D.F.); m.cordoni@studenti.unipi.it (M.C.); e.bechini@studenti.unipi.it (E.B.); antonio.darienzo@ao-pisa.toscana.it (A.D.); paolo.parchi@unipi.it (P.D.P.)

**Keywords:** aneurysmal bone cyst, pelvis, curettage, cryotherapy, pediatric orthopedics

## Abstract

Background: Aneurysmal bone cysts (ABCs) are benign but locally aggressive cystic lesions of the bone. Pelvic ABCs are extremely rare and hard to treat due to their high risk of local recurrence and the tough access to pelvic bones. Methods: In this retrospective study, we evaluated pediatric cases with pelvic ABC treated with curettage, cryotherapy and bone grafting treated in our institution between 2016 and 2022. Complications were recorded, as well as local recurrences. Patients’ post-operative functionality was assessed with the MSTS score. Results: Fourteen consecutive cases were included in our study. Their mean age at surgery was 13.5 years. The mean lesion size was 55 mm. The mean follow-up was 38 months. Two cases (11.8%) had local recurrences, which were successfully treated with further curettage. At their latest FU, 13 cases were continuously disease free (CDF), and one had no evidence of disease (NED). Only one case had a post-operative compilation (wound dehiscence). Patients’ mean post-operative MSTS score was 29.6. Conclusions: Pelvic ABCs are a challenge, even for the most experienced orthopedic surgeon. Our study suggests that the association of an accurate curettage, intraoperative cryotherapy and bone grafting can be a reliable and effective therapeutic option for large-sized ABCs of the pelvis.

## 1. Introduction

Aneurysmal bone cysts (ABCs) are rare but rapidly and locally aggressive benign tumors. ABCs occur more commonly in skeletally immature patients, particularly in the first and second decades of life [1]. According to the 2020 WHO Classifications of Tumors of Bone, ABCs are classified as osteoclastic giant cell-rich tumors and can be either primary lesions (about 70% of all cases) or secondary to other bone neoplasms (30%) [2,3,4]. Several studies have demonstrated the presence of recurrent chromosome translocations of the USP6 gene in primitive ABCs, thus confirming their neoplastic origin [3,5,6]. The most frequently affected sites are the metaphysis of the long bones, such as the humerus, femur and tibia, but ABCs may also rarely occur in the pelvis [3,4,6,7,8,9]. Currently, there is no consensus treatment for all ABCs. The treatment of every single lesion is based on patients’ age, lesions’ location and size and depends on the experience of the single surgeon and his or her team [10]. Over the few last decades, various treatments have been proposed and described. Although cases of spontaneous healing or healing after biopsy have been described [3], the medical literature agrees that most aneurysmal bone cysts require targeted treatments to control or eradicate the disease. Minimally invasive treatments, such as curopsy, the injection of sclerosing agents, radiofrequency thermal ablation (RFTA), selective arterial embolization (SAE) and the systemic use of bisphosphonates or denosumab, have often been used—either as exclusive or adjuvant treatment—especially at the most challenging surgical sites, including the pelvic region [1,3,10,11]. More invasive treatments, such as en bloc resections of the whole involved bone segment, can provide good results with meager recurrence rates but may lead to unsatisfactory functional outcomes [1]. For this reason, intralesional curettage and bone grafting represent the treatment of choice in most cases, as they dovetail a reasonable risk of local recurrence with good post-operative functional outcomes [3,10].

In particular, the management of pelvic ABCs is particularly complex because of the tridimensional anatomy of the pelvic region and its limited accessibility. Due to their relatively hidden location within the pelvic girdle, ABCs may grow significantly before becoming symptomatic. Extending to the endopelvic region, they can even cause abdominal and urinary symptoms [8,12]. Once in the operative theater, pelvic ABCs have a high risk of intraoperative bleeding, whose control can be challenging. The proximity to neurovascular structures, endopelvic organs and hip cartilage makes their treatment difficult, even for the most experienced surgeon [7,9,13].

Several adjuvants can be used to aid bone curettage in pelvic lesions to prevent local recurrences and the necessity of further surgical interventions in a complex anatomical segment such as the pelvic region. Among other local adjuvants, such as cement, phenol [14], alcoholization [15] and argon beam ablation [16], cryotherapy has been considered, for a long time, a promising adjuvant for curettage to minimize the risk of local recurrence in cases with locally aggressive bone tumors.

In this article, we report our experience with the treatment of pelvic aneurysmal bone cysts in children and adolescents using a combination of curettage, intraoperative cryotherapy and bone grafting.

## 2. Materials and Methods

This single-center retrospective study was performed following the ethical standards of the 1964 Declaration of Helsinki and its later amendments. All patients and their families gave their written consent. Our study consisted of a review of all the pediatric patients diagnosed with a primary aneurysmal bone cyst of the pelvis and treated surgically with curettage, surface cryotherapy and bone grafting in our institution between May 2015 and May 2022.

We collected personal data for each case, and only patients younger than 18 years of age were included in our study. Only cases with a histological diagnosis of aneurysmal bone cyst were included.

Each case underwent a careful evaluation of their pre-operative X rays, CT scans and MRI images in order to orientate diagnosis, guide the surgical planning and estimate the tumor size (Figure 1). A pre-operative diagnosis of ABCs was established with a CT-guided needle biopsy.

Imaging evidence was used to localize lesions in the pelvic bone, as cases were sorted using the Enneking–Dunham classification and evaluating their relationships with the surrounding bone and soft tissues according to the Capanna classification. The pre-operative performances of our patients’ lower limbs were assessed using the MSTS score. In our institution, curettage represented the definitive treatment of choice for pelvic ABCs. Infiltrative treatments with sclerosing agents were used in cases with active lesions and sub-optimal cortical coverage to minimize cyst growth and prevent further thinning of the cortical bone. Embolization was limited to a neoadjuvant role for large lesions with identifiable terminal feeding vessels. Our institution’s Department of Interventional Radiology performed the procedure and decided which artery to treat. Cases treated with pre-operative embolization received it between 24 and 72 h before curettage to minimize intraoperative bleeding.

Surgery consisted of an open approach to the involved segment of the pelvic bone. The modified Stoppa approach was used for lesions localized in the pubis, the superior pubic branch and the interior surface of the iliac bone. Ilio-femoral approaches were chosen for cysts involving the anterior and lateral surfaces of the iliac bone and periacetabular region. A posterior approach to the iliac crest and the sacroiliac joint was used for lesions in the posterior segments of the iliac crest. 

Curettage was performed using Volkmann bone curettes and high-speed burrs. The tissue material obtained during curettage was sent to our pathologist to confirm the diagnosis of ABC. The surfaces of the resulting cavity were treated with alcoholization and cryotherapy. Cryotherapy was performed using cryoprobes to freeze water or cryo-gel. During the freezing phase, two or three cryoprobes were set within the neo-cavity and activated to induce temperatures as low as −150 °C for five minutes. Ice balls were formed on each cryoprobe’s tip (Figure 2).

To maximize the effectiveness of cryotherapy, all the cavity surfaces should come into contact with ice. After five minutes, freezing was stopped, and the thawing phase began. Temperatures were increased using hot water, and cryoprobes were temporarily removed. Thaw continued until the ice had melted entirely. Cryoprobes were then repositioned to provide the maximal freezing effect in different areas, and a further freeze-and-thaw cycle was carried out similarly. The cavity was then definitively filled with bone chips or unitary bone allografts, depending on the lesion’s width and localization (Figure 3).

Post-operative follow-up consisted of serial office visits, clinical evaluations and post-operative X-rays and MRIs. Cases were routinely visited 1, 3 and 9 months after surgery, while subsequent visits were scheduled depending on the needs of every single individual. The MSTS scores of each case were calculated at their latest follow-up, according to the combination of data observed and reported by the patient.

Local recurrences, confirmed via imaging findings and histological analysis, were recorded. Each complication with grade II or higher, according to the Clavien–Dindo Classification, was also recorded.

## 3. Results

Fourteen consecutive pediatric patients with ABC of the pelvic bones treated in our institution with curettage, cryotherapy and bone grafting were included in our study. There were eight females and six males, with a mean age at surgery of 13.3 years (5–16). Among our 14 cases, 7 had lesions in their iliac bone (Type I according to the Enneking–Dunham classification), and 4 had cysts confined to the anterior pelvic ring (Enneking–Dunham Type III). The remaining three cases suffered from ABCs localized at least partially within the periacetabular region (two Enneking–Dunham Type I–II and one Type II–III).

Aneurysmal bone cysts were classified according to the Capanna classification: seven cysts belonged to class I, three belonged to class II, three were class III lesions and one last case had a class V lesion. 

According to pre-operative images, the mean major diameter of treated lesions was 53.2 mm (25–80). Six of our cases had previous intralesional infiltrations with lauromacrogol, vitamin C, methylprednisolone and bone marrow concentrate. Four cases underwent pre-operative embolization to minimize the risk of hemorrhage during the intervention. On average, embolization was performed 37.5 h before curettage.

Our patients’ mean pre-operative MSTS score was 24.5 (24–28). The mean MSTS score of cases with periacetabular lesions (Enneking I–II or II–III) was 22.3, whereas it was higher for cases confined to the iliac (Enneking I; 24.6) or the ischio-pubic segment (Enneking III; 26.0). Globally, the mean values of each item were as follows: pain 3.2, function 3.4, emotional acceptance 4.5, supports 4.7, walking 4.3 and gait 4.5. There was no statistical correlation between the lesion size and patients’ functionality, according to a Pearson correlation test (r = −0.201; *p* = 0.491).

None of our cases had major intraoperative complications. Our patients’ mean post-operative FU was 38.4 months (14–78). Only one patient (Case 13) suffered from a post-operative complication, particularly a wound dehiscence that could be treated successfully with surgical debridement. None of our cases suffered from deep infections, vessel or nerve injuries, mechanical failures or soft tissue damage attributable to cryotherapy.

One of our fourteen cases (7.1%) was diagnosed with a local recurrence, whereas the remaining thirteen (92.9%) patients were continuously disease-free through their post-operative follow-up. Case 6 had a local recurrence six months after surgery, which was treated with further curettage, cryotherapy and bone grafting. After this second surgical intervention, the patient had no evidence of disease at his latest clinical and imaging evaluations.

All patients had an improvement in their functional status after the intervention. Patients’ mean MSTS score improvement amounted to 5.1 (1–8), and our cases’ mean post-operative MSTS score was 29.6 (28–30) out of 30. The mean values of each item were as follows: pain 4.9, function 4.9, emotional acceptance 4.9, supports 5.0, walking 4.9 and gait 5.0. The mean post-operative MSTS score was significantly higher than that recorded before surgery, as testified via a two-tailed *t*-student test (*t* value = 9.38; *p* < 0.0001).

A schematic resume of our cases is reported in detail in Table 1.

## 4. Discussion

The treatment of ABCs arising from pelvic bones represents a challenge, even for the most experienced orthopedic oncologist. Once the diagnosis of aneurysmal bone cyst has been established, careful pre-operative planning should always be performed to choose the correct therapeutic approach for each case. Although intralesional injections or embolization can limit or even arrest the spread of some of these cystic lesions, open surgical approaches remain the best therapeutic option to eradicate the disease. In this regard, accurate curettage and bone grafting are reliable therapeutic options for pelvic ABCs in children and adolescents, as testified by the international literature [1,3,4,5,6,7,8,9,10,11,12,13]. In 1986, Capanna et al. [8] published their results on a cohort of 24 adult and pediatric cases with pelvic aneurysmal cysts. Eleven of these patients received curettage, and two had local recurrences, establishing a recurrence rate of 18%. In 2005, Mankin et al. [17] reported their twenty years of experience treating ABCs with curettage and bone grafting. Their population of patients of every age included 15 cases with lesions localized in the pelvis, with a local recurrence rate of 15%.

In the same year, a multicenter study by Cattalorda et al. [18] included 11 pediatric cases suffering from pelvic ABCs and treated with curettage. Their recurrence rate was 18% since two cases were diagnosed with secondary lesions, respectively, 6 and 13 months after surgery. In a later study, Novais et al. [13] reported their experience with 13 pediatric ABCs of the pelvis, who had a local recurrence rate as low as 7.7% (1 case out of 13). Finally, Deventer et al. [19] evaluated the clinical outcomes of six pediatric patients with pelvic ABCs treated with curettage in a recent paper. Among these six cases, three had consequential bone grafting, whereas the remaining three had PMMA. Three of the six young patients had a local recurrence after surgery.

Several adjuvant treatments have been proposed over the decades to minimize the risk of local recurrence [10]. Intraoperative cryotherapy is one of the most promising techniques associated with accurate curettage [20,21,22]. Cryo-induced tissue injuries have been proven to result from damage mechanisms during the freezing and thawing phases. These two consequential steps are responsible for cytologic and histologic alterations that could undermine cells’ homeostasis and tissue functionality. During the freezing phase, cold progressively compromises the inner and superficial cellular structures due to alterations in the three-dimensional architecture and the limitation of reciprocal movements of both proteins and lipids. Without sufficient thermal energy, enzymes undergo a dramatic downgrade in their performance. Membrane lipids lose a large share of their mobility, increasing the rigidity of the membranes. At the same time, ice crystals form in the extracellular spaces, causing a relative detriment in liquid water in the extracellular compartment. The resulting hyperosmotic extracellular environment draws water from the cells and stresses plasmatic membranes. As time passes and temperature decreases, the severe lack of intra-cellular liquid water, associated with an increased electrolyte concentration, is sufficient to induce a metabolic shock and destroy the cells.

Once the freezing phase ends, temperatures rise, and cells’ metabolic activity restarts. These events inevitably bring up the consequences of freezing-induced damage and alterations. During the thawing phase, cryo-mediated mitochondrial damage is thought to activate caspase cascades, forcing damaged cells to apoptosis. Melting ice makes the extracellular environment hypotonic; water enters the damaged cells, and their volume increases until their membranes break. Altogether, these mechanisms are theorized to induce damage in neoplastic cells while partially preserving the extracellular architecture of the surrounding healthy bone.

Furthermore, moving the point of view from the single cell to the tissue, the thawing phase is characterized by vascular dysfunctions caused by the rupture of some capillaries and the intraluminal stasis of larger vessels [23,24,25,26,27,28,29,30]. This extended vascular damage triggers a coagulative cascade that culminates in coagulative necrosis, which could be particularly effective in richly vascularized lesions such as aneurysmal bone cysts. In 1995, Marcove et al. [20] published their experience of 55 cases with ABCs localized in different anatomical districts treated with curettage and frozen liquid nitrogen, comparing them with 44 cases that received curettage alone. The latter group had a recurrence rate of 59%, while those who had cryosurgery had a risk as low as 18%. Further evidence was provided by Schreuder et al. [31] in 1997. These authors published a paper on 27 ABCs treated with curettage and cryotherapy, with only one local recurrence over a mean follow-up of 47 months. In 2009, Peeters et al. [32] reported a recurrence rate of 5% among their 80 cases of ABCs treated with curettage and cryotherapy administered with modern-designed cryoprobes. Our current study is consistent with what emerged from previous studies, providing a particular focus on the surgical treatment of pelvic lesions. The relatively low recurrence rate observed in our patients (7%), compared to those provided by previous studies (Table 2) [8,17,18,20,33,34,35,36,37,38,39,40], testifies to the effectiveness of our treatment.

In particular, our results suggest that cryotherapy is an effective local adjuvant therapy to clean up the bone surfaces after the curettage of ABCs. The combination of cryotherapy and curettage could make this latter an even more suitable option for the treatment of pelvic ABCs, compared to alternative strategies such as selective arterial embolization (SAE). In 2010, Rossi et al. [33] reported their experience using embolization as a primary treatment for 17 pelvic ABCs. Although embolization alone could avoid the bleeding associated with open surgical approaches, 47% of their cases with pelvic lesions required more than one embolization, and 29% had a local recurrence after their second embolization. These recurrence rates are higher than the one reported in our series, as a larger share of our cases did not require further surgical treatments or embolization. Early and complete resorption of the disease is desirable to preserve the treated bone and minimize the risk of pathologic fractures and acquired deformities in pediatric patients. Furthermore, despite the invasiveness of our approach, only one of our cases suffered from wound dehiscence, whereas none of our cases had major intraoperative complications.

Our cohort also confirms the low incidence of cryo-induced local complications, as none of our patients suffered from any damage directly attributable to the administration of cryotherapy.

From a functional point of view, our cases significantly improved their functional status after surgery. Their mean MSTS score rose from the 24.5 recorded before the treatment to a mean value of 29.6 at the patients’ latest follow-up. Our treatment was effective in restoring the performance of the pelvic region and providing pain relief, not only in relatively accessible areas, such as the pubis and the iliac crest, but also in districts that are difficult to reach, such as the periacetabular region. Our results are consistent with the ones obtained by Peeters et al. [32] and Novais et al. [13], thereby suggesting that the combination of curettage and cryotherapy can also lead to optimal outcomes for children and adolescents, who could live the rest of their lives without significant functional limitations.

We acknowledge that our study is not free of limitations. One of them is the retrospective nature of our study, which did not allow for the complete standardization of the post-operative follow-up procedures for each patient. As with most similar studies, our series also included lesions localized in different areas of the pelvic bone, which further reduced our grade of standardization. The small size of our cohort represents another limitation. The rarity of these tumors and the limited timespan of the investigation did not allow us to operate in a broader population, which partially limited the statistical significance of some of the data associations we wanted to investigate at the beginning of our research. These issues could be overcome in the future by performing similar evaluations on a prospective basis and broader populations.

Beyond these limitations, our study testifies to the effectiveness of a combination of curettage, cryotherapy and bone grafting in providing adequate local disease control and allowing for good functional outcomes in a mid-to-long-term scenario. These encouraging results, associated with low complication rates, could lead to greater use of cryotherapy as a local adjuvant treatment in pelvic aneurysmal bone cysts in children and adolescents.

## Figures and Tables

**Figure 1 healthcare-11-02658-f001:**
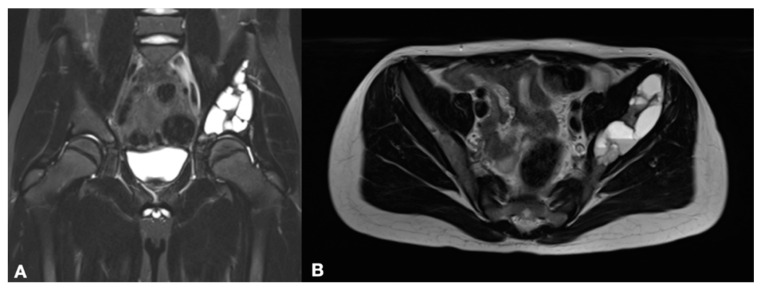
Coronal (**A**,**B**) transverse MRI images of a large ABC involving the left iliac bone and extending to the periacetabular region.

**Figure 2 healthcare-11-02658-f002:**
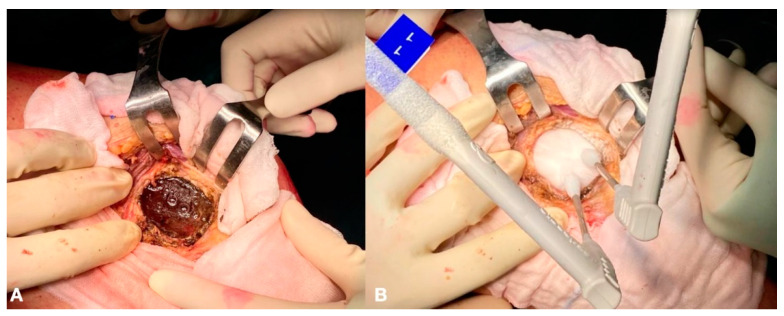
After an accurate curettage, the resulting cavity is filled with cryo-gel (**A**), which is frozen using two cryoprobes (**B**).

**Figure 3 healthcare-11-02658-f003:**
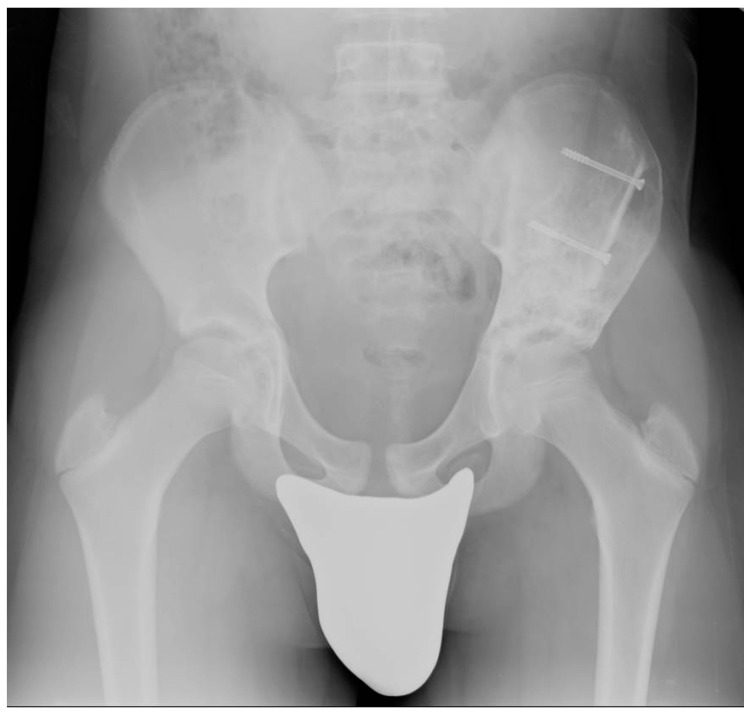
A post-operative X-ray image of the pelvis of a case who had suffered from an ABC involving the left ilium. The X-ray shows a good grade of osteointegration in the bone grafts one year after the intervention.

**Table 1 healthcare-11-02658-t001:** A resume of our cohort.

Case	Age	Size (mm)	Enn. Cl.	Cap. Cl.	Embol.	Infiltrat.	MSTS Pre-Op	Complic.	L.R.	Further Treatments	MSTS Post-Op	**F-U** **(m)**
1	5	25	I	I	No	No	23	-	No	-	30	58
2	15	55	I	I	No	No	27	-	No	-	30	36
3	16	35	I	III	No	Yes	24	-	No	-	30	28
4	14	45	I	I	No	No	27	-	No	-	30	78
5	12	65	I	III	No	Yes	24	-	No	-	30	16
6	12	65	I	II	No	Yes	24	-	Yes	Curettage + Cryotherapy+	28	68
									(6 M)	Bone grafting		
7	9	40	I	I	No	Yes	23	-	No	-	30	36
8	14	55	I–II	I	No	No	23	-	No	-	30	40
9	16	80	I–II	II	Yes	No	22	-	No	-	30	14
10	16	60	II–III	V	Yes	Yes	22	-	No	-	29	32
11	15	80	III	II	Yes	No	24	-	No	-	29	42
12	12	55	III	III	No	Yes	28	-	No	-	29	38
13	15	40	III	I	No	No	26	Dehiscence	No	Debridement	30	32
14	16	45	III	I	No	No	26	-	No	-	30	18

Enn. Cl. = Enneking-Dunham Classification; Cap. Cl = Capanna Classification; Embol. = Embolization; Infiltrat. = Infiltrations; Complic. = Complications; L.R. = Local Recurrence; F-U = Follow Up.

**Table 2 healthcare-11-02658-t002:** A brief summary of previous literature about the surgical treatment of pelvic ABCs.

Treatment	Article	Pelvic ABCs	Local Recurrence (%)	Complication Rate (%)	Follow-Up (Months)
**Curettage**					
Alone/High speed burr	Biesecker et al., 1970 [33]	7	59% *	-	-
Alone/High speed burr	Vergel De Dios et al., 1992 [34]	18	17%	-	-
Alone/High speed burr	Bollini et al., 1998 [35]	4	42% *	-	60 *
Alone/High speed burr	Gibbs et al., 1999 [36]	4	0%	-	87 *
Alone/High speed burr	Cattalorda et al., 2005 [18]	11	18%	0%	50
Alone/High speed burr	Mankin et al., 2005 [17]	13	20% *	-	72 *
Alone/PMMA	Ozaki et al., 1997 [37]	11	17% *	-	59
Alone/PMMA	Deventer et al., 2022 [19]	6	50%	17%	66
Alone/Phenol	Capanna et al., 1985 [8]	9	22%	22%	61
Argon beam coagulation	Steffner et al., 2011 [38]	6	17%	19% *	30 *
Cryotherapy	Marcove et al., 1995 [20]	<7	17% *	14% *	85 *
Cryotherapy	Schreuder et al., 1997 [31]	3	4% *	15% *	47 *
Cryotherapy	Peeters et al., 2009 [32]	9	5% *	6% *	55 *
Cryotherapy	Our Study	14	7%	7%	38
**Alternative strategies**					
Curopsy	Reddy et al., 2014 [39]	<9	19% *	-	14 *
Selective arterial embolization	Rossi et al., 2010 [40]	17	47%	5%	-

* the value refers to a population that does not include only pelvic lesions.

## Data Availability

The data that support the findings of this study are available from the corresponding author upon reasonable request.

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
