# Peer review of "Aneurysmal Bone Cyst of the Pelvis in Children and Adolescents: Effectiveness of Surgical Treatment with Curettage, Cryotherapy and Bone Grafting"

_healthcare, 2023, doi:10.3390/healthcare11192658_

Round 1
Reviewer 1 Report
The Authors aim to evaluate a series of pediatric cases with pelvic aneurysmal bone cyst treated with curettage, cryotherapy and bone grafting.
The topic is interesting.
Nonetheless, the paper presents major flaws.
Was IRB approval available?
The series is very small and extremely heterogeneous.
Large and small ABC, different areas of the pelvis are analyzed altogether. Minimum follow up to short to rule out a recurrence.
It is not clear whether the cases are 17 or 14..
Which are the potential advantages over selective arterial embolization? Surgery is deemed by high bleeding risk.
I would appreciate a table resuming different treatment of ABC, reporting success and complication rates.
Conclusions not supported by results.
Finally, the paper is disorganized and it presents many grammar and syntax errors.
the paper is disorganized and it presents many grammar and syntax errors.
Author Response
Dear Reviewer,
Thank you for your help and your suggestions to increase the quality of our paper.
We are pleased that the topic of our manuscript is to your liking. Our clinical practice has been performed in accordance with the ethical standards of the Declaration of Helsinki, and it was approved by our local Institutional Review Board (IRB).
We are conscious that the size, the heterogeneity, and the follow-up of our series represent a limit for the significance of our statistical analysis. In fact, reading the final part of our discussion, you could find that we had already identified the size and the limited timespan of investigation among the acknowledged limitations of our paper. We had not mentioned the fact that treated lesions were localized in different areas of the pelvic bone since most similar series in literature include lesions arising from ischium, ilium and pubis. However, following your suggestion, we have included in our revised manuscript the sentence “… our series also included lesions localized in different areas of the pelvic bone, which furtherly reduced our grade of standardization” (LINES 274-276).
By seeing through these limitations, we invite you to consider that modern literature lacks large scale studies on the topic. For example, the influencing article “Aneurysmal bone cysts of the pelvis in children: a multicenter study and literature review”, written by Cottalorda et al. in 2005, included only 11 cases (with lesions localized in ilium, ischium or pubis) treated with curettage.
Moving our attention to the combination of curettage and cryotherapy, Marcove et al., Schreuder et al. and Peeters et al. (cited in our discussion) had larger populations because they included cases with ABCs from all over the skeletal system, to the detriment of their degree of standardization. Still, none of these three papers included 14 or more cases of ABCs localized in the pelvis.
Our series included 14 cases. We are sorry for the mistake in LINE 14. It has been fixed.
As required, we introduced a new table (Table 2) to resume the results of the different surgical treatments for pelvic ABCs. Although the focus of our paper was on the curettage, which represents the first line treatment for these lesions, we have been pleased to mention also minimally invasive approaches. In particular, as you required, in LINES 249-262 we highlighted the potential advantages of the open surgical approach with curettage and cryotherapy compared to selective arterial embolization.
With regards to the scheme of narration, we appreciate your perspective, but we respectively disagree. Indeed, the introduction focused on ABCs and provided a first short overview of their possible treatments. Our discussion started providing an overview of the literature on curettage for pelvic ABCs and later introduced the rationale for the use of cryotherapy. Subsequently, our experience with cryotherapy and curettage for ABC was reported and compared with previous studies. Finally, we concluded that our functional and oncological results were encouraging, a statement supported by our results. The organization of our discussion and our conclusions did not represent a problem for the other reviewers, and we hope you could tolerate them.
Finally, we are sorry to receive your bad feedback about the quality of our English language, also considering that the other reviewers gave us significantly different feedbacks (minor or no editing required). We tried our best to fix the grammar and syntax errors with the help of a colleague who is an English native speaker.
We look forward to your response and we are available for any further change you might consider necessary.
Best regards,
E.I., M.D.

Reviewer 2 Report
The manuscript described the clinical outcome in patients with ABC at the pelvis.
The topic is interesting and some merit may be there in this manuscript.
Please confirm my comments.
1) How did the authors diagnose ABC before surgery? Biopsy? Some tumors look like ABC (ABC change).
2) Please show the mean score of each MSTS score such as pain and gait. The readers should have an interest in the improved score after surgery.
3) What was the indication of surgical intervention in this study?
4) Please add intraoperative bleeding because some patients underwent embolization.
5) The max. age was 16 years. I wonder if the title "children and adolescents" is appropriate.
No comments.
Author Response
Dear Reviewer,
Thank you for your help and your suggestions to increase the quality of our paper.
Please, find reported below our answers to your comments and suggestions:
- The diagnosis of ABC was established with pre-operative CT-guided needle biopsies and confirmed with further histological evaluations of the intralesional tissue obtained during curettage. The use of pre-operative needle biopsy has been included in our manuscript in LINES 81-82.
- The mean values for each item of the MSTS score have been included in our revised manuscript (LINES 152-153 for the pre-operative MSTS score and LINES 169-170 for the post-operative one)
- We provided a more extended and detailed description of the therapeutic approach to pelvic ABCs in our institution in LINES 90-95.
- Unfortunately, we did not record intra-operative blood loss, therefore we could not include this data.
- We are conscious that the World Health Organization (WHO) defines an adolescent as any person between ages 10 and 19, while none of our cases had more than 16 years of age. Despite the absence of cases aged 17 or 18 (that would have met our inclusion criteria in terms of age), we still decided to title our article "children and adolescents" since each of our cases was either a child or an adolescent. Obviously, we would consider changing this expression to others that could be more to your liking (“Pediatric patients?”).
We look forward to your response and we are available for any further change you might consider necessary.
Best regards,
E.I., M.D.

Reviewer 3 Report
An interesting clinical issue was well illustrated by the authors with a case cohort review. Some questions were raised as follows:
1. Reference No. 10 seems incomplete, should it be revised?
2. Were there some indications for embolization in this case cohort?
Author Response
Dear Reviewer,
Thank you for your help and your suggestions to increase the quality of our paper.
Please, find reported below our answers to your comments and suggestions:
- Reference number 10 was incomplete, as the title was missing. It has been fixed.
- We provided a more extended and detailed description of the therapeutic approach to pelvic ABCs in our institution (including embolization) in LINES 90-95
We look forward to your response and we are available for any further change you might consider necessary.
Best regards,
E.I., M.D.

Round 2
Reviewer 1 Report
Despite the Authors' efforts, the paper was not ameliorated enough.
Still many grammar errors
Data about pre-op embolization not precise.
Methods: the Authors declare approval by IRB. However, at the bottom, they acknowledged waived IRB approval. I would require IRB approval certification.
Also, a sub analysis of different pelvic areas would be an added value.
Many grammar errors.
Author Response
Dear Reviewer,
Thank you for your help and your suggestions to increase the quality of our paper.
We did our best to improve the quality of our English language further. We could find a few grammar errors that we fixed. We also tried to rephrase some of our sentences to make them more understandable and easy to read. We hope this version will be good enough. Otherwise, please point out some errors that should be fixed, and the phrases should be changed to better comply with your requests. English corrections are colored in dark red.
We provided further information about the timing between embolization and curettage (LINES 94-95 and LINE 152). These are all the data at our disposal, considering that these procedures were performed by a different team (as stated in LINES 93-94). Concurrently, we kindly invite you to consider that the embolization was not one of the main topics of our study and our manuscript since they were focused on the results of open surgery, particularly curettage.
We could not provide an IRB approval certification. For this reason, we will remove the sentence in the first paragraph of the materials and methods section and maintain the one in our IRB statement (LINES 297-298). We are sorry for the inconvenience.
We have been asked for a sub-analysis of different pelvic areas. The different surgical approaches for the different pelvic areas had already been described in LINES 97-102, and lesions were already categorized according to the Enneking – Dunham classification in LINES 138-142. In the results section of the latest version of our manuscript, we included the mean MSTS score of cases with lesions involving the periacetabular region, the iliac bone, and the ischio-pubic area (LINES 152-154). We also added the sentence, “Our treatment was effective in restoring the performance of the pelvic region and providing pain relief not only in relatively accessible areas, such as the pubis and the iliac crest but also in districts that are difficult to reach, such as the periacetabular region.” In the Discussion section (LINES 270-273).
Please find the changes other than language corrections colored in light red.
We look forward to your response and we are available for any further change you might consider necessary.
Best regards
Reviewer 2 Report
The authors reply accurately.
Author Response
Dear Reviewer,
Thank you for your help and suggestions to increase our paper's quality.
We are pleased that our replies and manuscript have been of your liking.
Best regards